# Attentional Functioning in Healthy Older Adults and aMCI Patients: Results from the Attention Network Test with a Focus on Sex Differences

**DOI:** 10.3390/brainsci15070770

**Published:** 2025-07-19

**Authors:** Laura Facci, Laura Sandrini, Gabriella Bottini

**Affiliations:** 1Department of Brain and Behavioral Sciences, University of Pavia, 27100 Pavia, Italy; laura.sandrini@unipv.it (L.S.); gabriella.bottini@unipv.it (G.B.); 2Cognitive Neuropsychology Centre, ASST Grande Ospedale Metropolitano Niguarda, 20172 Milan, Italy; 3NeuroMI, Milan Center for Neuroscience, 20126 Milan, Italy

**Keywords:** Mild Cognitive Impairment, Attention Network Test, attentional subcomponents, executive control, sex differences, gender medicine, precision medicine, cognitive aging

## Abstract

**Background/Objectives**: The prognostic uncertainty of Mild Cognitive Impairment (MCI) imposes comprehensive neuropsychological evaluations beyond mere memory assessment. However, previous investigations into other cognitive domains, such as attention, have yielded divergent findings. Furthermore, while evidence suggests the presence of sex differences across the spectrum of dementia-related conditions, no study has systematically explored attentional disparities between genders within this context. The current study aims to investigate differences in the attentional subcomponents, i.e., alerting, orienting, and executive control, between patients with MCI and healthy older controls (HOCs), emphasizing interactions between biological sex and cognitive impairment. **Methods**: Thirty-six participants (18 MCI, and 18 HOCs) were evaluated using the Attention Network Test (ANT). Raw RTs as well as RTs corrected for general slowing were analyzed using Generalized Mixed Models. **Results**: Both health status and sex influenced ANT performance, when considering raw RTs. Nevertheless, after adjusting for the baseline processing speed, the effect of cognitive impairment was no longer evident in men, while it persisted in women, suggesting specific vulnerabilities in females not attributable to general slowing nor to the MCI diagnosis. Moreover, women appeared significantly slower and less accurate when dealing with conflicting information. Orienting and alerting did not differ between groups. **Conclusions**: To the best of our knowledge, this is the first study investigating sex differences in attentional subcomponents in the aging population. Our results suggest that previously reported inconsistencies about the decline of attentional subcomponents may be attributable to such diversities. Systematically addressing sex differences in cognitive decline appears pivotal for informing the development of precision medicine approaches.

## 1. Introduction

According to current neuropsychological models, the attention system includes three primary subcomponents: alerting, orienting, and executive control [1,2,3,4]. Alerting refers to the ability to maintain a state of readiness and respond promptly to external cues; orienting involves selecting specific information among various sensory inputs and directing attention to it; executive control pertains to the monitoring and resolution of conflicts between competing stimuli, particularly in the context of decision-making, planning, and error detection [2]. It has been demonstrated that these attentional subcomponents can deteriorate independently across the lifespan, and their disruption may serve as an indicator of early stages of cognitive decline [5,6]. One widely used task for investigating attentional abilities and relative subcomponents, and tracking their changes in aging, is the Attention Network Test (ANT; [2,7,8]). ANT is a behavioral task specifically designed to assess the efficiency of the three attentional subcomponents by measuring reaction times (RTs) across different cue and flanker conditions. Moreover, from a neurofunctional perspective, it has been demonstrated that, during the execution of the ANT, three distinct neural networks are independently recruited, each supporting its corresponding subcomponent ([8]). However, several studies employing the ANT to assess performance in young healthy individuals, elderly adults, and patients with Mild Cognitive Impairment (MCI) or dementia have yielded inconsistent results (for a review: [4,6]). For instance, Fernández and colleagues [9] found a difference between healthy individuals and patients with MCI in the orienting subcomponent, only in the case of vascular MCI. Alterations in orienting abilities have also been reported when comparing patients with Alzheimer’s Disease (AD) with both young adults [10,11] and older adults [12]. Conversely, other studies observed selective impairments in the executive control among patients with MCI [5,13,14], or with AD [15], while no differences emerged in the orienting network. Notably, these alterations have not always been consistently replicated, with some authors even reporting no significant attentional differences between patients and healthy subjects [16,17]. A primary concern in interpreting previous findings lies in the lack of a systematic and consistent methodology across studies to control for general slowing. Indeed, Gamboz and colleagues [18] observed age-related differences in executive control among healthy individuals. However, when the overall response speed was taken into account, the magnitude of the conflict effect was substantially reduced, suggesting that the observed differences might be attributed to general slowing rather than to a specific executive dysfunction. Notably, alerting is the only subcomponent that appears to be consistently affected by aging, regardless of adjustments for general slowing [10,15,18,19,20,21,22]. However, whether and how alerting is further altered in the earliest stages of cognitive decline remains unclear. Similarly, given the heterogeneity in previous findings, it is still challenging to determine which specific alteration observed at the ANT may serve as a prodromal marker of cognitive decline. Interestingly, limited attention has been paid to factors beyond general slowing that might explain attentional performance in older adults. Specifically, emerging evidence points to sex-specific trajectories in cerebral and cognitive aging, with females showing increased amyloid-beta deposition and reduced hippocampal volume relative to males [23]. Moreover, despite women generally outperforming men in verbal memory tasks across both healthy and populations with MCI [24,25], this cognitive advantage tends to diminish with the onset of dementia, leading to a more rapid cognitive decline in women, who constitute approximately two-thirds of individuals diagnosed with AD [25,26]. The present study aims to investigate the role of sex in age-related changes in attentional functioning, by examining differences in the ANT performance between patients with amnestic MCI (aMCI) and healthy older adults. The purpose is twofold: first, to replicate previous findings on selective attentional deficits associated with prodromal stages of dementia—specifically MCI—while accounting for general cognitive slowing; and, secondly, to examine sex as a critical variable that may contribute to a deeper understanding of the mechanisms underlying cognitive changes in the progression from healthy to pathological aging.

## 2. Materials and Methods

### 2.1. Participants

A total of 36 participants were recruited for the study, including 18 patients diagnosed with single- or multiple-domain aMCI (10 females, 8 males; mean age: 77.8 ± 5.57; mean education level: 12.3 ± 4.27), and 18 healthy older adults (10 females, 8 males; mean age: 74.8 ± 5.81; mean education level: 14.1 ± 3.99) matched to the patient group for gender, age [*t* (34) = −1.58, *p* = 0.123, *d* = −0.53], and education level [*t* (34) = 1.29, *p* = 0.206, *d* = 0.43].

Experimental procedures were carried out at the Center of Cognitive Neuropsychology of the ASST Grande Ospedale Metropolitano Niguarda (Milan, Italy), and at the Department of Brain and Behavioral Sciences (DBBS) of the University of Pavia.

The study adhered to the ethical principles outlined in the Declaration of Helsinki (1964) and complied with Good Clinical Practice (GCP) guidelines. Ethical approval was obtained from the Ethics Committee of Milan Area 393 (Approval Code: 161-08032022) for the patient group, as part of a larger project assessing the validity of a cognitive stimulation program for MCI patients, and from the Ethics Committee of the University of Pavia for the healthy adults (Approval Code: 149/23). All participants provided their written informed consent before taking part in the study.

#### 2.1.1. Patients with aMCI (MCI)

The study was proposed to all patients who consecutively and autonomously accessed the Center of Cognitive Neuropsychology of Niguarda Hospital (Milan) for a neuropsychological evaluation from October 2022 to October 2024 and received a diagnosis of aMCI, or single- or multiple-domain, according to Petersen criteria [27,28,29].

Specifically, we selected patients with a Mini Mental State Examination (MMSE, [30]) corrected score ≥ 23, daily activities indices (ADL/IADL, [31] within the normal range, and a deficit score in at least the Free and Cued Selective Reminding Memory Test—Image version [32,33]. The diagnosis had to precede the experimental assessment by a maximum of six months.

Group-level neuropsychological assessment scores are reported in Table 1.

Exclusion criteria included the following: (i) the presence of psychiatric or neurological disorders accounting for the observed cognitive impairment; (ii) current or past substance abuse; and (iii) untreated or worsening primary sensory deficits (i.e., uncorrected vision or hearing impairment).

A total of 18 patients [10 females (mean age: 78.2 ± 5.79; mean education level: 11.4 ± 3.60), and 8 males (mean age: 77.4 ± 5.63; mean education level: 13.4 ± 5.01)] signed the informed consent and were enrolled in the study.

#### 2.1.2. Healthy Older Controls (HOCs)

We recruited 18 healthy volunteers, age-, gender-, and education-matched to the patients [10 females (mean age: 74.8 ± 4.78; mean education level: 14 ± 4.03) and 8 males (mean age: 74.9 ± 7.26; mean education level: 14.1 ± 4.22)]. Volunteers were recruited through the distribution of informational flyers at the University of the Third Age in Pavia (Italy) and provided written informed consent.

Exclusion criteria included the following: (i) age below 60 or above 90 years; (ii) diagnosed psychiatric or neurological disorders; (ii) current or past substance abuse; (iii) untreated or worsening primary sensory deficits, and (iv) performance below the cut-off on either the Montreal Cognitive Assessment (MoCA; [43]; Italian version: [44]) or the Free and Cued Selective Reminding Test—Picture version [32,33]. Group-level cognitive performance scores are presented in Table 2.

### 2.2. Experimental Procedure

Prior to testing, demographic and clinical data were collected for all participants.

Healthy controls underwent the cognitive screening session, including the MoCA and the FCSRT (~30 min), to confirm the absence of cognitive impairment.

After confirming eligibility, all participants performed the Attention Network Test (ANT) as part of the experimental protocol.

#### The Attention Network Test (ANT)

The ANT was administered on a MacBook Pro using the Psychology Experiment Building Language (PEBL) software suite, version 2.1. Task instructions were translated into Italian and simplified for clarity, and the stimuli were presented at a screen resolution of 840 × 525 pixels.

The task consisted of 24 practice trials, followed by three experimental blocks of 96 trials each, for a total of 288 trials. Each trial lasted 4000 milliseconds (ms) and followed a standardized sequence (See Figure 1).

First, a fixation cross appeared at the center of the screen for a variable duration ranging from 400 to 1600 ms. This was followed by a warning cue presented for 100 ms. The cue could be one of four types: no cue (no warning), central cue (an asterisk appearing at the fixation point), double cue (asterisks simultaneously presented at both possible target locations, above and below fixation), or spatial cue (an asterisk at the actual target location). After the cue, a short fixation interval of 400 ms preceded the target stimulus, which consisted of a central arrow flanked by either neutral condition (no flankers), congruent flankers (arrows pointing in the same direction as the target), or incongruent flankers (arrows pointing in the opposite direction). The target remained on screen until the participant responded, but for no longer than 1700 ms. Finally, a post-target fixation period was presented, calculated as 3500 ms minus the sum of the initial fixation duration and the participant’s RTs, ensuring that each trial lasted exactly 4000 ms [8].

Participants were instructed to determine the direction of the central arrow as quickly and accurately as possible by pressing either the left or the right shift key.

Upon task completion, the PEBL software automatically generates a TextEdit summary report, including the total number of errors, mean accuracy, and mean RTs.

### 2.3. Statistical Analysis

Statistical analyses were performed using Jamovi software (version 2.4.7, Intel, https://www.jamovi.org/).

RTs above 1700 ms and incorrect trials were excluded from the dataset. Then, mean RTs for each cue type (no cue, double cue, central cue, and spatial cue), combined in the congruent, incongruent, and neutral conditions, were considered as main variables (see Descriptives in Table 3).

Since RT distributions were positively skewed and violated the assumption of normality, as confirmed by the Shapiro–Wilk test (*p* < 0.001), generalized linear mixed models (GLMMs) were employed. GLMMs were specified using a Gamma distribution with an identity link function. The dependent variable was the mean RTs, with fixed effects for Health condition (MCI vs. HOCs), Sex (male vs. female), Cue type, and Flanker Congruency. To control for potential speed–accuracy trade-offs, the number of errors per each condition was included as a covariate. Participant ID was modeled as a random intercept to account for within-subject variability. The confidence interval for all parameter estimates was set at 95%.

Given the previously discussed effect of general cognitive slowing, an additional analysis was conducted on proportional RT scores to account for individual differences in baseline processing speed. For each participant, the mean RTs for each condition was divided by their overall mean RTs [18]. These proportional scores were then z-transformed across the sample to standardize the data, enabling between-subject comparisons while accounting for individual variability in processing speed. A mixed model, appropriate for standardized variables containing both positive and negative values, was applied using the same structure as the original GLMM. Because of the limited sample size, model estimation was performed using restricted maximum likelihood (REML) and *p*-values were derived using the Satterthwaite approximations for degrees of freedom [45,46]. Model residuals confirmed the assumption of normality (*p* = 0.662).

While GLMMs allow independent evaluation of the effects of single-task conditions, they are not designed to capture composite indices of attentional functioning, i.e., alerting, orienting, and executive control, which are traditionally computed as difference values between specific cue and flanker types (see Figure 1). To address this issue and facilitate comparison with previous studies, the alerting, orienting, and executive control components were, here, calculated following the method proposed by Festa-Martino and colleagues [10]. Specifically, for each network, an index was computed by subtracting the proportional RTs of each relevant cue or flanker condition [8,10]. Then, three separate repeated-measures ANOVAs were run with indices as dependent variables, and Health Condition and Sex as between-subject factors. All post hoc comparisons were Bonferroni-corrected for multiple testing. The threshold for statistical significance was set at *p* < 0.05.

## 3. Results

### 3.1. Raw RTs

The GLMM conducted on the mean RTs revealed a significant main effect of Health Condition (X^2^_[1]_ = 117.42; *p* < 0.001), confirming that patients are significantly slower than HOCs (see Figure 2a) (β = 153.93 ms, *z* = 10.84, *p* < 0.001).

Furthermore, consistent with the previous literature, significant main effects of Flanker Congruency (X^2^_[2]_ = 903.28; *p* < 0.001), Cue Type (X^2^_[3]_ = 59.90; *p* < 0.001), as well as a significant interaction between Flanker Congruency and Cue Type (X^2^_[6]_ = 30.56; *p* < 0.001), were observed. Post hoc comparisons indicated that all participants, regardless of health condition and sex, responded slower in incongruent trials than in congruent ones (Δ = 111.7 ms, *z* = 19.3, *p* < 0.001), as expected when conflicting information is provided. With regard to cue types, participants responded faster both in the spatial cue condition compared to the central cue condition (Δ = 19.42 ms, *z* = 3.16, *p* = 0.010), demonstrating an orienting effect, and in the double cue condition compared to the no cue condition (Δ = 34.08 ms, *z* = 5.39, *p* < 0.001), consistent with an alerting effect.

Additionally, a significant main effect of Sex (X^2^_[1]_ = 53.39; *p* < 0.001) indicated that males are generally faster than females (see Figure 2b) (β = −99.38 ms, *z* = −7.31, *p* < 0.001). However, a significant Health Condition*Sex interaction (X^2^_[1]_ = 36.13; *p* < 0.001) revealed that, while healthy males and females show comparable performance (Δ = 17.5 ms, *z* = 0.921, *p* = 1.000), females with aMCI are significantly slower than both healthy males (Δ = 253.3 ms, *z* = 12.67, *p* < 0.001) and males with aMCI (Δ = 181.3 ms, *z* = 9.29, *p* < 0.001). Moreover, healthy males are faster than males with aMCI (Δ = −72 ms, *z* = −3.55, *p* = 0.002; see Figure 2c). A full description of the results of the GLMM is documented in the Appendix A.

It is important to note, however, that these results are based on raw RTs and do not account for individual baseline speed differences. To control for such inter-individual variability, additional analyses were conducted on proportional and standardized RTs scores.

### 3.2. Z-Scores

The mixed-effect model conducted on z-scored RTs confirmed the significant main effects of Health Condition (F_[1,27.6]_ = 4.35; *p* = 0.046), Flanker Congruency (F_[2,351.9]_ = 647.42; *p* < 0.001), Cue Type (F_[3,344.2]_ = 28.37; *p* < 0.001), and Sex (F_[1,25]_ = 5.17; *p* = 0.032), as well as significant Flanker Congruency*Cue Type (F_[6,344.2]_ = 7.36; *p* < 0.001) and Health Condition*Sex interactions (F_[1,25.1]_ = 4.95; *p* = 0.035).

Focusing on the latter interaction, a post hoc simple effects analysis showed that the effect of Health Condition is significant only among females (F_[1,27.6]_ = 9.55; *p* = 0.005), but not among males (F_[1,25.5]_ = 0.002; *p* = 0.966). Specifically, when accounting for general slowing, females with aMCI perform significantly worse than their healthy peers (β = −0.22 ms, *t*_[27.6]_ = −3.09, *p* = 0.005), whereas males with aMCI and healthy males show comparable performance (β = −0.003 ms, *t*_[25.5]_ = −0.04, *p* = 0.966). Z-scores are shown in Figure 3a.

Moreover, in contrast with the analysis on raw data, an additional significant main effect of Errors was found (F_[1,47]_ = 104.34; *p* < 0.001), along with a significant interaction between Sex and Flanker Congruency (F_[2,347.6]_ = 4.14; *p* = 0.017). Specifically, a post hoc simple effects analysis revealed that the effect of Sex is only significant in incongruent flanker trials (F_[1,154]_ = 13.14; *p* < 0.001), indicating that differences between males and females emerge solely under conditions of cognitive conflict (β = 0.30 ms, *t*_[154]_ = 3.63, *p* < 0.001). No significant sex differences were observed in congruent trials (β = 0.04 ms, *t*_[160]_ = 0.47, *p* = 0.642). A post hoc comparison is shown in Figure 3b. A full description of the results of the mixed model is documented in the Appendix A.

### 3.3. Attentional Network Indices

To further investigate the indices of attentional functioning, we conducted three separate repeated-measures ANOVAs using the indices of alerting, orienting, and executive control calculated from the proportional scores [10].

For both the alerting and orienting indices, there were no significant main effects of Health Condition and Sex, nor any significant interaction between Health Condition*Sex (all *p_bonf_* > 0.06). These results are in line with the GLMM analysis, showing comparable performance in alerting and orienting cues between groups.

In contrast, a significant main effect of Sex was observed for the executive control index (F_[1,32]_ = 4.66, *p* = 0.039, η^2^ = 0.116), confirming sex-related differences in the management of conflicting information. No significant effects were found for Health Condition, nor for the Health Condition*Sex interaction (all *p_bonf_* > 0.06). These results are consistent with the mixed-effect model findings, where a significant Sex*Congruency interaction emerged.

Interestingly, the post hoc analysis showed that women exhibit a significantly smaller difference in response time between incongruent and congruent conditions (Δ females = 90.92 ms; Δ males = 122.18), compared to men (*p_bonf_* = 0.039). This suggests that, despite being generally slower than men (females: M = 897.46 ms ± 181.43; males: M = 772.02 ms ± 101.53), and specifically on both incongruent (females: M = 933.54 ms ± 152.66; males: M = 855.69 ± 93.83 ms) and congruent trials (females: M = 842.62 ± 141; males: M = 733.51 ± 80.26), when faced with conflicting information, women appear faster in resolving the conflict. A full description of the results of the ANOVA is documented in the Appendix A.

Although this result, adjusted for differences in slowing, suggests a better conflict resolution in the female population, we argue that accuracy must be considered, as it may have influenced this outcome. Thus, we run an error rate analysis on the executive control index.

A repeated-measures ANOVA was conducted, with the mean number of errors in congruent and incongruent flanker trials as the 2-level within-subject factor, and Sex and Health Condition as the grouping variables. Descriptive statistics for proportional RTs and error rates are provided in Table 4.

The analysis revealed a significant main effect of Flanker Congruency (F_[1,32]_ = 8.12, *p* = 0.008, η^2^ = 0.009), meaning that participants appear more accurate in congruent with respect to incongruent trials (Δ = −3.14, *t*_[32]_ = −2.85, *p_bonf =_* 0.008). Moreover, a main effect of Health Condition (F_[1,32]_ = 10.58, *p* = 0.003, η^2^ = 0.091) indicate that healthy participants are overall more accurate than patients (Δ = −9.89, *t*_[32]_ = −3.25, *p_bonf =_* 0.003). Notably, an interaction between Flanker Congruency*Sex*Health Conditions emerged (F_[1,32]_ = 4.95, *p* = 0.033, η^2^ = 0.006), showing a significant difference in accuracy on incongruent trials between patients and controls, but only among females (Δ = −17.80 ms, *t*_[32]_ = −3.56, *p_bonf =_* 0.033). In contrast, no significant difference in error rates was observed between HOCs and MCI males on the same trials (*p_bonf =_* 1.0). This three-way interaction is shown in Figure 4. Extended results are reported in the Appendix A.

## 4. Discussion

Although it is well-established that attentional networks play a crucial role in supporting a wide range of high-level cognitive functions, their trajectory during the lifespan and their alterations in both healthy and pathological aging remain not fully understood. Here, we aimed to investigate the functioning of attentional subcomponents in healthy older adults and patients with aMCI, arguing that inconsistencies in previous findings might not be solely attributable to methodological or statistical artifacts (i.e., differences in applying correction for general slowing). Specifically, biological sex has received limited attention in prior studies, despite increasing recognition of its potential role in cognitive aging and decline. To the best of our knowledge, no previous study has investigated the impact of sex on attentional functioning in the context of aMCI.

Given the established utility of the ANT in detecting changes in attentional functioning and its potential role as a diagnostic tool, we administered it to older adults with and without aMCI, ensuring that groups were matched for age, gender, and education level. To avoid biased comparisons between groups, we first analyzed raw RTs and then applied z-score transformations to proportional data, thereby standardizing performance after correcting for individual differences in baseline processing speed.

Initial analyses on uncorrected RTs revealed that participants with aMCI are significantly slower than healthy controls across all task conditions, consistent with the expected pattern of general slowing in MCI. All participants, regardless of health status or sex, demonstrated alerting, orienting, and executive control abilities. Specifically, participants responded more rapidly in the double cue compared to the no cue condition, indicating increased vigilance in response to preparatory signals. They also responded faster to spatial cues relative to central cues, suggesting the effective orienting of attention when spatial information was available. Lastly, a significant difference between congruent and incongruent conditions reflected the presence of a conflict-resolution effort. Interestingly, a main effect of the variable sex suggests that males are generally faster than females; moreover, a significant interaction between sex and health condition indicates that this difference is evident only within the MCI group.

This result becomes more pronounced and takes a slightly different form after correcting and standardizing RTs for individual baseline speed. Indeed, among male participants, the differences between MCI patients and controls were no longer significant after accounting for the baseline speed, suggesting that the observed deficits in men with aMCI may be largely attributable to general slowing. In contrast, among women, attentional deficits persisted after correction, with female participants with aMCI exhibiting slower RTs and a lower accuracy compared to healthy women and both males’ groups. This finding suggests that attentional deficits in women with aMCI may not be solely caused by general slowing.

A similar trend emerged under the condition of cognitive conflict, regardless of diagnosis, with women appearing slower than men in responding to incongruent trials. The performance at the congruent trials was, instead, equivalent between men and women. In order to properly understand the meaning of this difference, we run an ANOVA applying the definition of executive control as formulated by Fan and colleagues [7], i.e., calculating an index of the difference in RTs between congruent and incongruent trials. Larger discrepancies in such executive control index indicate a greater cognitive effort in resolving the conflict. Interestingly, women showed a smaller executive control, suggesting a smaller interference effect than men in conflicting situations, independently from the diagnosis. However, it is important to note that, despite this smaller absolute difference, women generally remain slower than men in overall reaction times. Moreover, while the standard definition of conflict resolution in the ANT does not include error rates, we argue that accuracy must play a role in task performance. Ultimately, even in real-life contexts, a conflict can only be considered truly resolved if it is resolved correctly, not merely quickly.

Accordingly, we also examined the accuracy differences between men and women across both congruent and incongruent trials. The results showed that, although all women appear to exhibit a reduced conflict effect, females with aMCI also demonstrate a significantly lower accuracy when dealing with contrasting information.

Interestingly, previous studies on functional abilities have shown that performance accuracy significantly declines over time in individuals with MCI, but not in cognitively healthy peers. Specifically, patients with MCI tend to maintain their speed on tasks at the expense of accuracy, reflecting a maladaptive Speed–Accuracy Trade-off (SAT; [47]). Our study confirms this dysfunctional SAT, but only among females with aMCI, who show a reduced response accuracy in incongruent trials, as well as overall slower performance at the ANT. Therefore, it is plausible that sex may also influence compensatory mechanisms in MCI.

To summarize the results, a difference in general slowing was observed between patients and individuals with aMCI, as expected. Once general slowing was controlled for, no significant differences emerged in alerting or orienting between patients and controls, nor between men and women. We speculate that the lack of alerting differences may reflect its age-related decline, as reported in previous studies comparing younger and older adults [10,15,18,19,20,21,22]. Given that our sample includes only older individuals, alerting might be uniformly reduced, thus not constituting a marker of aMCI. However, significant differences were found in both overall reaction times and executive control, with women—and, in particular, women with aMCI—showing a reduced effectiveness in maintaining executive control. These findings align with previous evidence indicating selective impairment in the conflict index among patients with MCI [5,13,14] and further suggest that sex may be a key factor driving this difference.

We speculate that these processes are potentially shaped by underlying risk factors. First, endocrine processes appear to be closely linked to neuroinflammatory mechanisms, with estrogens playing a particularly important role in suppressing microglia activation, reducing pro-inflammatory cytokine production, and promoting neurotrophic factors [48,49]. Estrogen levels decline sharply during menopause, whereas men experience a more gradual reduction in androgens, potentially resulting in a lower exposure to inflammatory burden [50]. Additionally, neuroimaging studies have further corroborated sex differences, showing that post-menopausal women exhibit greater amyloid accumulation mainly in the frontal brain regions [51], and hypometabolism in temporo-parietal areas compared to age-matched men and pre-menopausal women [52]. Notably, other studies have shown that men with AD often present with more extensive brain atrophy compared to women, despite similar clinical symptoms [53,54]. This has been interpreted as a greater neural reserve in men, which may delay the emergence of cognitive symptoms [53].

Finally, psychological and sociocultural factors have also been suggested to exacerbate women’s vulnerability to cognitive decline [25]. Particularly, a historically reduced educational background, limited healthcare access, a higher burden in caregiving roles, and, overall, greater chronic stress exposure may collectively heighten women’s risk [25,55]. Moreover, depression has been consistently associated with an increased risk of AD, and women exhibit twice the lifetime prevalence of depressive symptoms compared to men [55].

Future research should aim to replicate and expand our findings in larger and more diverse populations, while accounting for biological, psychosocial, and environmental factors known to influence vulnerability and resilience in cognitive aging.

Importantly, we acknowledge that the relatively small sample size represents the main limitation of the present study. This constraint was due to our recruitment strategy, which involved the consecutive enrollment of patients based on their access to the hospital. Thus, the findings of this study should be interpreted as indicative of a trend that requires confirmation in larger samples.

The direction of such a trend aligns with the recent evidence on sex differences in cognitive decline. Accordingly, our study further emphasizes that sex is not a secondary variable but, rather, may play a role in how cognitive decline manifests and is detected. We argue that inconsistencies in previous findings on attentional subcomponents in both healthy and pathological aging might, at least partially, be explained by sex differences.

Although much research on cognitive decline has focused on memory impairment, we decided to center our study on attention. This choice was informed by the recent evidence suggesting that a more comprehensive neuropsychological assessment, including the evaluation not only of memory but also attention, executive functions, language, and visuospatial abilities, can be highly valuable for both differential diagnosis and for monitoring the progression of the disease [29,56,57,58]. Additionally, longitudinal studies combining a neuropsychological assessment with postmortem analyses have shown that attention is often the first cognitive domain to decline, even before episodic memory, in asymptomatic patients with AD neuropathology [59]. Therefore, refining the understanding of changes at the level of attentional subcomponents may help in further improving our comprehension of the mechanisms underlying different aging trajectories.

This is the first study that investigated sex differences in attentional subcomponents in healthy elderly and aMCI patients. Further studies on larger samples may have direct clinical implications. The available diagnostic tools, in fact, do not account for sex-specific cognitive profiles with the risk of underestimating impairment in women or misinterpreting physiological age-related patterns.

In conclusion, this study contributes valuable insights to further the advancement of precision medicine [60]. Indeed, our findings highlight that sex may influence one’s susceptibility to attentional alterations and should, therefore, be systematically considered as an individual factor when developing personalized preventive and therapeutic strategies.

## Figures and Tables

**Figure 1 brainsci-15-00770-f001:**
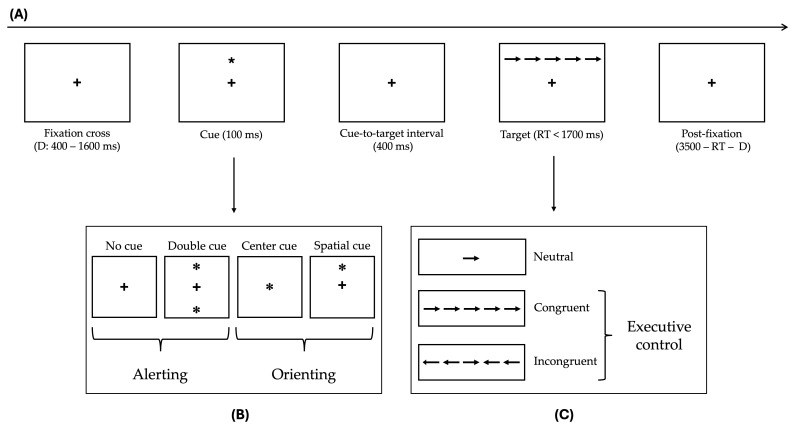
Schematic representation of the ANT experimental procedure. (**A**) Timeline showing the sequence of events of a typical trial, with a total duration of 4000 ms. (**B**) The four cue conditions, used to compute alerting (RTs no cue—RTs double cue) and orienting (RTs central cue—RTs spatial cue) indices. (**C**) The different flanker types, whose contrast enables the estimation of executive control efficiency.

**Figure 2 brainsci-15-00770-f002:**
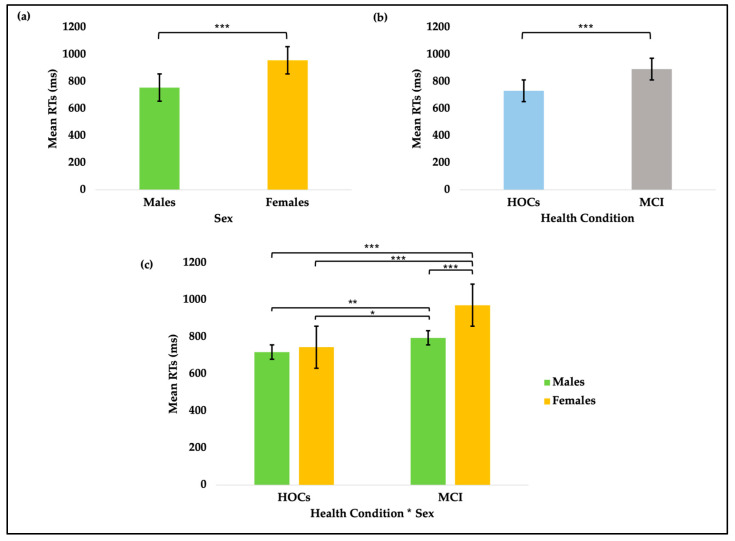
Bar plots of mean raw RTs in milliseconds. Error bars represent 95% confidence intervals. Bonferroni post hoc; * *p* < 0.05; ** *p* < 0.01; *** *p* < 0.001. (**a**) Bars represent average RTs by Health Condition: MCI patients are significantly slower than HOCs (*p* < 0.001). (**b**) Bars show average RTs by Sex: males are significantly faster than females (*p* < 0.001). (**c**) Bars represent average RTs as a function of Health Condition and Sex. Females with aMCI are significantly slower compared to all groups (*p* < 0.001). Males with aMCI are slower than healthy males (*p* = 0.002) and healthy females (*p* = 0.029). No other pairwise contrasts reached significance.

**Figure 3 brainsci-15-00770-f003:**
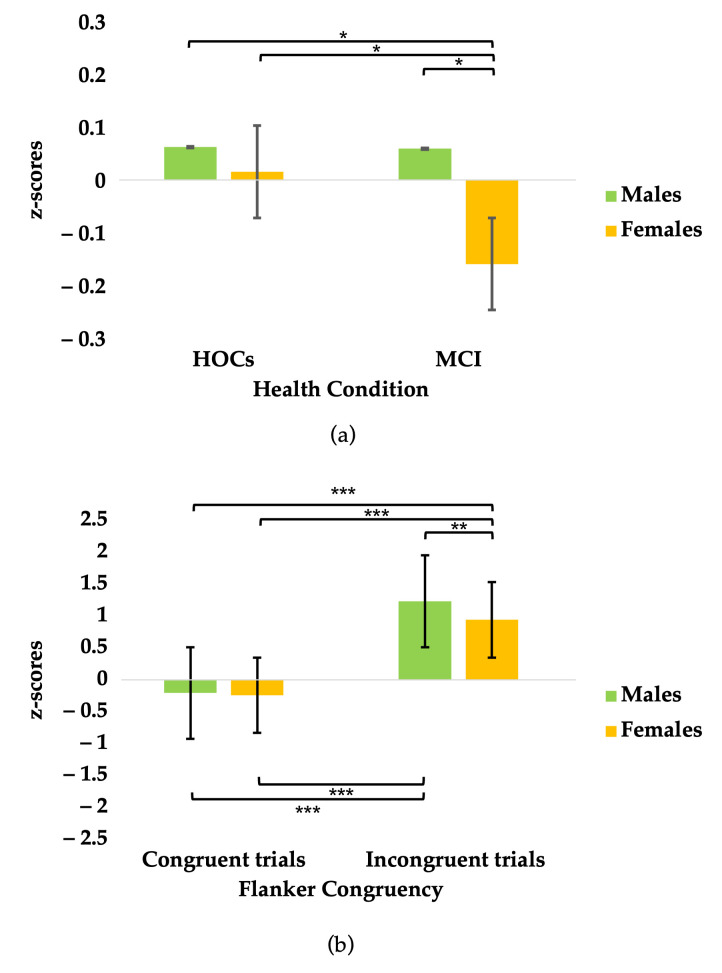
Results of mixed model analyses on standardized RT z-scores. Error bars denote 95% confidence intervals. Bonferroni post hoc; * *p* < 0.05; ** *p* < 0.01; *** *p* < 0.001. (**a**) z-scores by Health Condition and Sex. Bars represent estimated marginal means of z-scores across all task conditions, adjusted for individual baseline speed. (**b**) Z-scores by Flanker Congruency and Sex. Bars represent estimated marginal means of z-scores across all cue conditions.

**Figure 4 brainsci-15-00770-f004:**
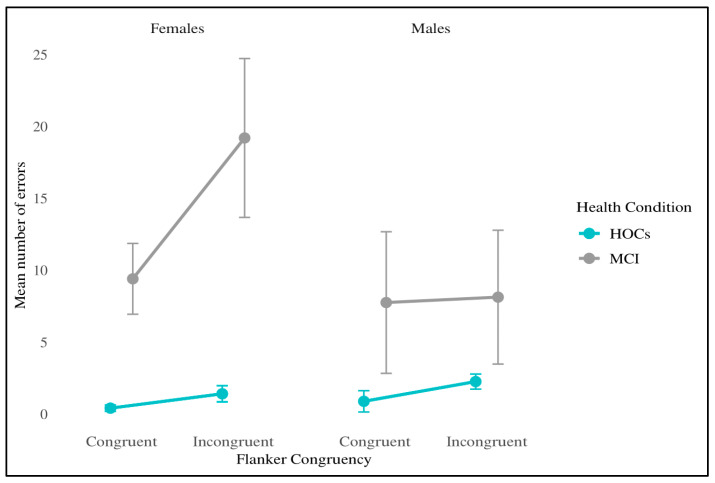
Error rates as a function of Health Condition, Sex, and Flanker Congruency. Line plots show the average number of errors in congruent and incongruent trials across groups. Error bars denote 95% confidence intervals. No significant group differences were found for congruent trials. In incongruent trials, females with aMCI exhibited significantly higher error rates compared to all other groups.

**Table 1 brainsci-15-00770-t001:** MCI patients’ scores at the neuropsychological assessment. Scores are adjusted based on Italian normative data. MMSE, Mini Mental State Examination; ADL/IADL, Activities of Daily Living/Instrumental Activities of Daily Living; FCSRT, Free and Cued Selective Reminding Memory Test; MTCF, Modified Taylor Complex Figure; TMT, Trail Making Test; SDMT, Symbol Digit Modality Test; FAB, Frontal Assessment Battery. () maximum score for the test when available.

TEST	Normative Cut-Off	MCI (*n* = 18) Adjusted Score
General cognitive status		
MMSE [30]	>23.86	25.92 ± 1.55 (30)
ADL [31]	>5	5.94 ± 0.24 (6)
IADL [31]	>7	7.41 ± 0.87 (8)
Language		
Verbal fluency [34]	>17.77	32.88 ± 10.33
Semantic fluency [34]	>28.34	36.01 ± 7.42
Visual naming test [35]	-	z score 4.53 ± 4.17
Verbal memory		
FCSRT [33]		
- Immediate Free Recall	>21.26	12.52 ± 5.46 (48)
- Delayed Free Recall	>8.1	3.57 ± 3.11 (16)
Digit span [36]	>4.26	5.94 ± 1.07 (9)
Digit span backward [36]	>2.65	4.95 ± 0.89 (8)
Visuo-spatial memory		
MTCF [37]		
- Direct copy	>27.66	32.04 ± 4.46 (36)
- Delayed recall	>8.4	8.04 ± 3.77 (36)
Corsi span [36]	>3.46	5.59 ± 0.85 (9)
Corsi span backward [36]	>3.08	5.1 ± 0.76 (8)
Attention and executive functions		
TMT [38]		
- A	<93	39.9 ± 16.25
- B	<282	110.45 ± 66.32
Stroop Test [39]		
- Time	<36.92	20.58 ± 15.55
- Errors	<4.24	0.38 ± 1.22
SDMT [40]	>34.2	44.4 ± 9.23 (110)
FAB [41]	>13.4	16.8 ± 3.8 (18)
Abstract reasoning		
Raven’s colored matrices [42]	>17.50	34.06 ± 2.4 (36)

**Table 2 brainsci-15-00770-t002:** Healthy older adults’ scores at neuropsychological tests. Scores are adjusted based on Italian normative data. MoCA, Montreal Cognitive Assessment; FCSRT, Free and Cued Selective Reminding Memory Test. () maximum score for the test when available.

TEST	Normative Cut-Off	HOCs (*n* = 18) Adjusted Score
MoCA [44]	>17.362	24.23 ± 1.48 (30)
FCSRT [33]		
- Immediate Free Recall	>19.59	30.24 ± 3.53 (36)
- Delayed Free Recall	>6.31	10.72 ± 1.24 (12)

**Table 3 brainsci-15-00770-t003:** Descriptive statistics of mean RTs from the ANT. Mean (and SD) of average RTs calculated over 288 trials per participant, stratified by group (MCI vs. HOCs), sex (female vs. male), flanker type (congruent, incongruent, and neutral), and cue type (no cue, double cue, central cue, and spatial cue).

			Cue Type			
Condition	Sex	Flanker Type	No Cue	Double Cue	Central Cue	Spatial Cue
MCI	Females	Congruent	971 (101)	941 (119)	982 (101)	949 (118)
		Incongruent	1080 (114)	1061 (117)	1074 (122)	1043 (103)
		Neutral	906 (80)	869 (93)	873 (112)	907 (111)
	Males	Congruent	812 (49)	729 (53)	780 (70)	749 (87)
		Incongruent	908 (56)	933 (107)	895 (72)	854 (50)
		Neutral	775 (88)	693 (82)	692 (82)	718 (95)
HOCs	Females	Congruent	758 (85)	725 (53)	724 (56)	709 (73)
		Incongruent	836 (116)	845 (89)	844 (90)	794 (82)
		Neutral	705 (69)	668 (51)	666 (29)	659 (59)
	Males	Congruent	752 (115)	690 (79)	692 (95)	664 (78)
		Incongruent	816 (103)	831 (111)	824 (96)	788 (109)
		Neutral	692 (88)	619 (69)	632 (99)	618 (90)

**Table 4 brainsci-15-00770-t004:** Descriptive statistics of executive control. Mean (and SD) of raw RTs, proportional scores, executive control index, and number of errors. Data are categorized by group (MCI vs. HOCs), sex (female vs. male), and flanker type (congruent and incongruent).

			Executive Control Index			
			RTs			
Condition	Sex	Flanker Type	Mean RTs	Proportional RTs	Conflict Index	Number of Errors
MCI	Females	Congruent	956 (101)	0.92 (0.04)	0.08 (0.11)	9.4 (7.8)
		Incongruent	1032 (149)	1.0 (0.15)		19.2 (17.5)
	Males	Congruent	768 (57)	0.94 (0.04)	0.16 (0.02)	7.75 (14)
		Incongruent	897 (66)	1.1 (0.05)		8.13 (13.2)
HOCs	Females	Congruent	729 (56)	0.97 (0.02)	0.14 (0.04)	0.4 (0.7)
		Incongruent	835 (75)	1.11 (0.02)		1.4 (1.78)
	Males	Congruent	699 (89)	0.97 (0.02)	0.16 (0.05)	0.88 (2.1)
		Incongruent	814 (103)	1.12 (0.03)		2.25 (1.49)

## Data Availability

The raw data supporting the conclusions of this article will be made available by the authors upon request.

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
