# Peer review of "Attentional Functioning in Healthy Older Adults and aMCI Patients: Results from the Attention Network Test with a Focus on Sex Differences"

_brainsci, 2025, doi:10.3390/brainsci15070770_

Round 1

Reviewer 1 Report

Comments and Suggestions for Authors

Dear Author,

Thank you for the opportunity to review your manuscript “Attentional functioning in healthy older adults and aMCI patients: results from the Attention Network Test with a focus on gender”

The objective of this manuscript was to “ investigate differences in the attentional sub- components, i.e., alerting, orienting, and executive control, between patients with MCI and healthy older controls (HOCs), emphasizing interactions between gender and cognitive impairment”

As the paper is focused on gender, there is little or any reference to the evaluation of gender as the use of sex is predominant within the manuscript.

Line 10 you state” The prognostic uncertainty of Mild Cognitive Impairment (MCI) imposes comprehensive neuropsychological evaluations beyond mere memory assessment.” Please explain what this sentence means for what is imposed and what are neuropsychological evaluations??

Within line 13-15 you use the term Sex and Gender, however, they have not been quantified or explained what you mean by these terms.??

Line 28 you state” Systematically addressing sex differences in cognitive decline appears pivotal for advancements in precision medicine.” Why is this your summary statement when the manuscript is a focus on gender? Please explain what is precision medicine and why is this important.

Line 35 why is the attention model different from the other models and if this is the case then is the attention model the primary focus of the manuscript??

Line 42 -85 there are any topics introduced that do not relate to the topic and may be considered within the discussion and not in the introduction,  There is a need to focus on the topic of ANT , why it is used, how important it is, why is sex different to gender and what components you are assessing and the gaps in the research.

It is noted in line 86 that the aim of the manuscript has changed from gender to sex, please explain as this does not set the scene for confidence in the rest of the manuscript.

“The present study aims to investigate the role of sex in age-related changes in attentional functioning, by examining differences in the ANT performance between patients with amnestic MCI (aMCI) and healthy older adults”.

Within the materials there is no G Power assessment to indicate the numbers of participants required to gain a confidence level. This overall is a very small number of participants.

Line 114 you state” The study was proposed to consecutive patients who autonomously accessed the Center of Cognitive Neuropsychology of Niguarda Hospital (Milan) for a neuropsychological assessment from October 2022 to October 2024 and received a diagnosis of aMCI, 116 single- or multiple-domain, according to Petersen criteria [32].” Please explain consecutive patient ? participants, why is the Petersen criteria used and how validated is the assessment.

Line 355 please explain graceful and pathological aging remain controversial.??

Overall, the study needs to be reconstructed and if gender is the focus then why is it not highlighted throughout the manuscript.

Prior to a final review the document needs to be re-considered with the initial aim of the manuscript either considered or re-evaluated.

Comments on the Quality of English Language

Please have this read by a native english speaker prior to re-submission

Reviewer 2 Report

Comments and Suggestions for Authors

This manuscript reports on a behavioural experiment aimed at assessing attentional components (alerting, orienting, and conflict/executive functions) by means of the Attention Network Task (ANT) in patients with mild cognitive impairment (MCI), with a specific focus on sex differences.

The topic is timely and relevant. As correctly noted by the authors in the introduction, the literature on ANT is mixed, with different papers highlighting different attentional components as being specifically affected by physiological and/or pathological aging. Moreover, the manuscript is clear and well written.

That said, I am not fully convinced that the current sample size is large enough to draw firm conclusions, rather than simply adding further variability to an already mixed body of literature on this topic. The manuscript currently does not offer any rationale for the sample size used. No power analyses (either a priori or post hoc) are provided, making it difficult to assess whether the experiment is adequately powered. This aspect is particularly concerning given that the study also focuses on sex differences, effectively splitting both the patient and control groups into two. In fact, there are only eight male subjects per group (and female groups are not much larger), which is a very small sample size by current standards.

Furthermore, there are substantial reasons why this comment on sample size is not merely “formal” but also critical. Several null effects are discussed in the manuscript (to give just one example from the abstract: “the effect of cognitive impairment was no longer evident in men”, lines 21–22). As is well known, absence of evidence does not constitute evidence of absence. The authors should therefore clearly state the rationale for the chosen sample size and provide evidence that their analyses have sufficient statistical power.

One possible solution, in the case of low statistical power, could be the use of Bayesian statistics, which can help determine whether there is substantial evidence supporting either the presence or absence of effects.

Additionally, recent studies investigating attentional components with the ANT in physiological aging (PMID: 34413509; PMID: 39851374) have suggested that a more fruitful approach might be parametric/correlational—i.e., examining variation in alerting, orienting, and executive control as a function of age. I wonder whether a similar approach might be applicable here, not in terms of age, but rather in relation to the severity of clinical symptomatology. For instance, observing that conflict scores vary as a function of clinical severity would certainly strengthen the current analysis.

Finally, the “conceptual” approach to the analysis is not entirely clear to me. I commend the authors for analysing RT data using linear mixed models, and I understand that this approach may not be directly applicable to the computation of the classical ANT indices (alerting, orienting, and conflict), which require subtraction methods. However, I also wonder how the current results can be properly compared with the existing ANT literature, which typically includes these indices. I would therefore suggest conducting a separate analysis using the classical subtraction method to calculate alerting, orienting, and conflict effects, and to show more explicitly how these vary across groups and subgroups.

Reviewer 3 Report

Comments and Suggestions for Authors

The manuscript is well-structured and presents a compelling rationale, clearly identifying gaps in the current literature on attentional functioning in the prodromal stages of cognitive decline, with a novel focus on sex differences (please use sex I think is more appropriate. See also latest publications by Boccardi V et al on this field to improve introduction and discussion). The methodology is rigorous and appropriately detailed, with careful control of confounding variables and the application of Generalized Linear Mixed Models and standardized z-score transformations enhances the robustness of the statistical analysis. However, the results section could benefit from greater conciseness; it is at times overly detailed with numerical data, which may detract from readability and hinder interpretative clarity. Moreover, the claim that women exhibit superior executive control—despite higher error rates—should be interpreted with more caution. This apparent “efficiency” might reflect a speed–accuracy trade-off rather than a  cognitive advantage. The discussion could further elaborate on this tension and emphasize the need to jointly consider both response time and accuracy. While the discussion is thorough and well-referenced, the interpretation of sex-specific vulnerabilities in relation to hormonal and psychosocial factors remains somewhat speculative; integrating more concrete evidence from neuroimaging or neuroendocrinology would strengthen these claims. Overall, the paper represents a valuable and original contribution to the neuropsychology of aging, with important implications for sex-sensitive diagnostics and precision medicine. Nonetheless, the impact of the findings would be enhanced by more cautious interpretation and tighter synthesis of key results.

Round 2

Reviewer 1 Report

Comments and Suggestions for Authors

Dear Authors,

Thank you for the update and the explanation.  It is appreciated you have enhanced your manuscript.

Reviewer 2 Report

Comments and Suggestions for Authors

The authors have adequately addressed my concerns.